# SELF-PREFERENCE BIAS IN LLM-AS-A-JUDGE

## ABSTRACT

Automated evaluation leveraging large language models (LLMs), commonly referred to as LLM evaluators or LLM-as-a-judge, has been widely used in measuring the performance of dialogue systems. However, the self-preference bias in LLMs has posed significant risks, including promoting specific styles or policies intrinsic to the LLMs. Despite the importance of this issue, there is a lack of established methods to measure the self-preference bias quantitatively, and its underlying causes are poorly understood. In this paper, we introduce a novel quantitative metric to measure the self-preference bias. Our experimental results demonstrate that GPT-4 exhibits a significant degree of self-preference bias. To explore the causes, we hypothesize that LLMs may favor outputs that are more familiar to them, as indicated by lower perplexity. We analyze the relationship between LLM evaluations and the perplexities of outputs. Our findings reveal that LLMs assign significantly higher evaluations to outputs with lower perplexity than human evaluators, regardless of whether the outputs were self-generated. This suggests that the essence of the bias lies in perplexity and that the self-preference bias exists because LLMs prefer texts more familiar to them.

## 1 INTRODUCTION

Measuring the quality of responses in dialogue systems presents unique challenges due to the diverse range of response strategies. Nowadays, thanks to the strong text understanding of large language models (LLMs), various automatic evaluations can be easily implemented by employing LLMs as evaluators (often referred to as LLM evaluators or LLM-as-a-judge). MT-Bench (Zheng et al., 2024) is an example of a benchmark utilizing this approach to score dialogue systems.

However, as Deutsch et al. (2022) reported, LLM evaluators are inherently biased. The inherent bias might cause inappropriate testing and inject biased preferences into the target dialogue systems (Zheng et al., 2024). One of the most significant biases is self-preference bias, which refers to the tendency of LLMs to overestimate the quality of their own outputs, as demonstrated in Figure 1a. The self-preference bias poses potential risks, including the promotion of specific ideologies or response styles intrinsic to the LLM evaluator.

Several studies have addressed the issue of self-preference bias; however, there is a lack of reliable metrics to quantify the extent of self-preference bias, and the fundamental causes of this phenomenon remain unclear. Koo et al. (2024) proposed a benchmark incorporating self-preference bias. Panickssery et al. (2024) investigated its relationship with self-recognition capabilities. However, neither study has quantitatively assessed self-preference bias while considering its disparity from human evaluations. Xu et al. (2024) and Stureborg et al. (2024) addressed quantifying self-preference bias within an evaluation approach where LLMs assign an absolute score to a single generated text. In this approach, gold-standard evaluators are required to assign scores based on abstract criteria while ensuring consistency with prior evaluations, making it challenging to obtain accurate assessments. As a result, these studies often restricted their scope to specific tasks, such as text summarization or machine translation, and relied on reference-based metrics like BLEURT (Sellam et al., 2020), which does not reflect the diversity of real-world use cases.

By contrast, a pairwise evaluation approach that involves direct comparison between two texts enables evaluators to recognize specific differences more readily, resulting in more consistent human judgments. Consequently, such pairwise evaluation methods are particularly suitable for analyzing biases related to discrepancies with human evaluations.

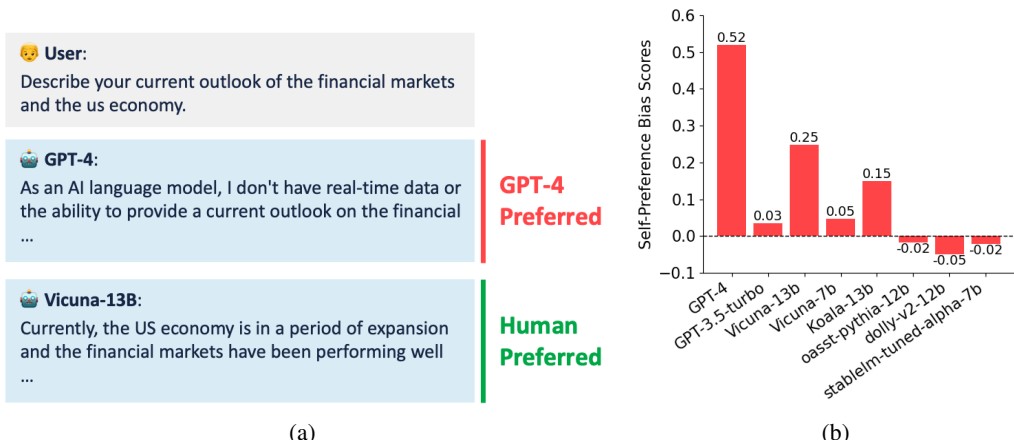

(a)                                          (b)

Figure 1: **How much do LLMs prefer their own responses over human evaluations?** (a) illustrates an example where GPT-4 favors its own response, even when human evaluations prefer to a response generated by Vicuna-13B. (b) compares the self-preference bias scores using our proposed metric (Definition 4.1). These figures demonstrate that GPT-4 exhibits a stronger self-preference bias than other models, suggesting that it tends to rate its own outputs more favorably than human evaluations. For detailed experimental settings, refer to Section 4.

In this paper, we measure the self-preference biases of LLMs in the pairwise evaluation. To accomplish this, we propose a new metric to quantify self-preference bias on the basis of algorithmic fairness concepts, thereby enabling discussions within the existing frameworks of fairness. In our experiment, we measured self-preference bias in eight LLMs. The results indicated that GPT-4 exhibited a significant self-preference bias (Figure 1b). This finding suggests a potential concern: using GPT-4 as a judge may lead to excessive influence from GPT-4's unique styles and policies.

Furthermore, we investigated the underlying causes of self-preference bias. Although LLM evaluators are not explicitly informed whether a given text is their own, they still exhibit self-preference bias. We hypothesized that LLM evaluators might be affected by the perplexity of the text, which tends to be lower perplexity when it is generated by themselves. To test this hypothesis, we analyzed the relationship between the perplexities of the texts to be evaluated and their corresponding evaluations. Our analysis revealed that LLMs assign significantly higher evaluations to texts with lower perplexity than human evaluators, regardless of whether the texts were self-generated. This suggests that the fundamental cause of self-preference bias may be the familiarity of the texts to the LLM evaluators, specifically how likely they are to generate the same response.

**Contributions** The contributions of this paper are threefold: (1) We propose a new metric to quantify self-preference bias in LLMs; (2) Using this metric, we evaluate the extent of self-preference bias across eight different LLMs; and (3) We identify a tendency for LLMs to assign higher ratings to texts with lower perplexity while exploring potential causes for this self-preference bias.

## 2 RELATED WORK

LLM-as-a-judge has been studied in various directions. The use of large language models (LLMs) for benchmarking has been advancing steadily. LLMs are increasingly employed in dialogue evaluation due to their ability to provide flexible assessments from perspectives such as utility and safety (Zheng et al., 2024; Sottana et al., 2023; Wang et al., 2024; Schick et al., 2021). Furthermore, numerous studies have explored the use of feedback from LLMs to enhance the learning of the LLMs themselves (Madaan et al., 2023; Chen et al., 2024). In particular, pairs of texts annotated with their relative quality have been proven to be useful training signals (Sun et al., 2023; Yuan et al., 2024), combined with various learning methods (Ouyang et al., 2022; Rafailov et al., 2023).

The self-preference bias is only one of several limitations inherent in the LLM-as-a-judge paradigm. Zheng et al. (2024) identified other biases such as position bias, where specific positions within a prompt are preferentially selected, and verbosity bias, which favors longer responses. Position bias can be addressed relatively easily by rearranging the order of comparative options within the prompt, and alignment-based methods to mitigate the bias have also been proposed (Li et al., 2024). Additionally, Saito et al. (2023) investigated the presence of verbosity bias in GPT-4 and GPT-3.5-Turbo, focusing on the quantification of bias by examining the error rates associated with longer versus shorter text options.

## 3 PRELIMINARIES: FAIRNESS AND BIAS

The measurement of bias in classifiers has been widely discussed within the framework of fairness (Calders et al., 2009; Hardt et al., 2016). In this section, we focus on a representative definition of fairness, Equal Opportunity, as the preparation for quantifying the self-preference bias in LLM evaluators.

Equal Opportunity (Hardt et al., 2016) is a fairness definition that requires the classifier to achieve equal recall across groups with different sensitive attributes (e.g. gender or race). Specifically, let the sensitive attribute be $S \in \{0, 1\}$, the prediction of the classifier be $Y' \in \{0, 1\}$, and the ground truth label be $Y \in \{0, 1\}$. The classifier satisfies Equal Opportunity if the following condition holds:

$$P(Y' = 1 | S = 1, Y = 1) = P(Y' = 1 | S = 0, Y = 1) \qquad (1)$$

When quantifying bias in the classifier, the difference or ratio between both sides of the equation are often used. The amount of bias derived from this definition is the difference in how well the classifier matches the ground truth between groups with different sensitive attributes. Therefore, if there is already bias towards sensitive attributes during the creation of the ground truth, Equation 1 cannot be considered an ideal definition of fairness.

To address this drawback, the fairness definition known as Demographic Parity (Calders et al., 2009), which does not rely on ground truth, is also widely used. Demographic Parity requires that the predictive distribution of the classifier be consistent across groups with different sensitive attributes. It is based on the assumption that there is no inherent causal relationship between the sensitive attributes and the predicted labels.

LLM-as-a-judge is a technique aimed at replacing human evaluators with LLMs, and more specifically, it assumes that dialogue system are aligned based on human preferences. Therefore, in this study, we employ the concept of Equal Opportunity to quantify self-preference bias by treating LLM evaluators as classifiers.

## 4 QUANTIFYING SELF-PREFERENCE BIAS

In this section, we propose a new metric to quantify self-preference bias. In particular, we focus on a setting where evaluators compare two texts and select the one higher quality, allowing comparison with reliable human evaluations.

### 4.1 SELF-PREFERENCE BIAS METRIC

To measure the extent to which an LLM's evaluation deviates from human evaluations across its own responses and those generated by others, we employ the concept of Equal Opportunity (Hardt et al., 2016). The bias is quantified by calculating the difference between both sides of Equation 1. A rigorous definition is provided below.

**Definition 4.1.** *(self-preference bias of the evaluator $f$.) Let $f$ be the evaluator assessing the quality of dialogue responses, capable of generating its own responses. For a pair of responses $y_0$ and $y_1$ in a dialogue, define: $Y \in {0, 1}$ as the index of the human-preferred response, $Y' \in {0, 1}$ as the index of the $f$-preferred response, and $S \in {0, 1}$ as the index of the $f$-generated response. We define the self-preference bias of the evaluator $f$ in dialogue comparison evaluation as follows:*

$$Bias = P(Y' = 1 | S = 1, Y = 1) - P(Y' = 1 | S = 0, Y = 1). \qquad (2)$$

In this definition, the amount of bias is represented by the difference between the conditional probability of the evaluator $f$ rating itself favorably given that the human evaluator has rated it favorably, and the conditional probability of the evaluator $f$ rating itself unfavorably given that the human evaluator has rated it unfavorably. A value of $0$ indicates the absence of bias, while a value close to $1$ suggests a high degree of bias. Conversely, a value of $-1$ would indicate the presence of a reverse bias, where the evaluator $f$ tends to undervalue its own responses. In this definition, the case where $y_1$ is preferred is explicitly considered, but this does not result in a loss of generality by treating the preferred response as $y_1$.

## 4.2 EXPERIMENTAL SETTING

The aim of our experiment is to quantify the self-preference bias in various LLMs. To address a wide range of tasks and topics, we have LLMs evaluate responses in open-ended dialogues and measure the bias using the Definition 4.1.

First, we provide the LLM evaluator with a user query written by a human and two responses generated by different LLMs. We then ask the LLM evaluator to determine which of the two responses demonstrates higher quality. For clarity and ease of reference during evaluation, we designate the two responses as "response A" and "response B". Finally, using the probabilities that the LLM evaluator outputs for "A" and "B", denoted as $p(\text{A}|\text{context})$ and $p(\text{B}|\text{context})$, we calculate the score for response A using the following equation.

$$\text{score}_\text{A} = \frac{p(\text{A}|\text{context})}{\sum_{w \in \{\text{A, B}\}} p(w|\text{context})} \tag{3}$$

The score for response B is calculated in the same manner. This post-processing, as shown in Equation 3 is designed to enhance the interpretability of analyses across LLMs with different probability distributions, following the approach of Schick et al. (2021). Additionally, according to Zheng et al. (2024), LLMs may exhibit position bias, which refers to the tendency to prefer responses located in specific positions within the prompt. To mitigate this bias, we swap the positions of responses A and B, and the evaluation scores are averaged over two iterations. Using the scores obtained through the above process, we examine the extent to which there is a difference in evaluation scores between the LLM's own generated response and responses generated by other LLMs.

The experimental setup may include biases other than position bias, such as verbosity bias. Position bias stems from the design of the experimental setup, whereas other biases are intrinsic to the model. To comprehensively measure self-preference bias specific to the model, we have not attempted to mitigate biases other than position bias.

Empirical evaluation uses Chatbot Arena dataset (Zheng et al., 2024), which contains 33,000 dialogues and each consisting of a user query and a pair of responses generated by two different LLMs. We calculate evaluation scores for the pre-existing response pairs stored in the dataset. In this dataset, every response pair is labeled with either "model_a", "model_b", or "tie" as a result of a human evaluation comparing the quality of the responses.

As the LLM evaluators, we employed the following eight LLMs, which also used in Chatbot Arena dataset: the closed-source GPT-3.5-Turbo and GPT-4 (Josh et al., 2024), and the open-source Vicuna-7b, Vicuna-13b (Chiang et al., 2023), oasst-pythia-12b (Biderman et al., 2023), dolly-v2-12b (Conover et al., 2023), Koala-13b (Geng et al., 2023), and stablelm-tuned-alpha-7b (Jonathan Tow, 2023). We used the same prompt as Zheng et al. (2024) to calculate the evaluation scores of responses A and B for all LLM evaluators. The prompt instructs the LLM to output either "[[A]]", "[[B]]", or "[[C]]" after providing an explanation. In the implementation, we used the output probability distribution following the token corresponding to "[[" in the generated text to compute the score using Equation 3. We excluded 1.07% of the responses that failed to comply with the prompt instructions, such as not providing a response in the required format (e.g., "[[A]]" or a similar structure). The temperature value used in the experiment was set to 0.7, following the reference provided in Zheng et al. (2024).

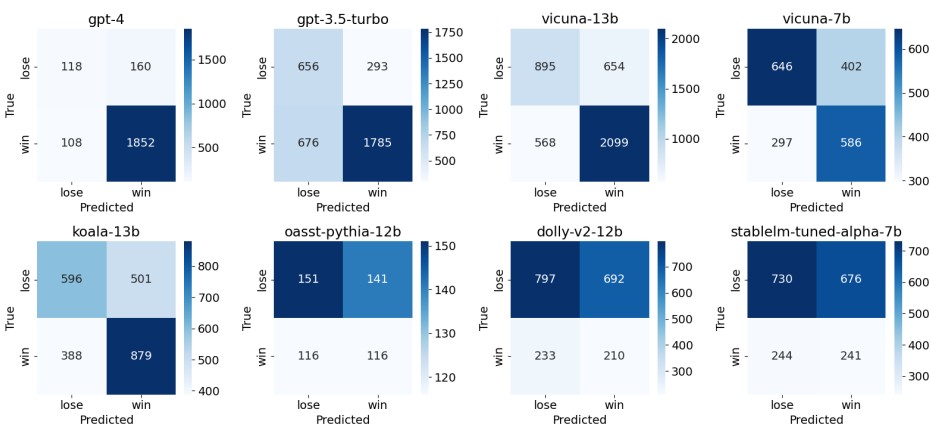

Figure 2: Confusion matrix for each LLM evaluator's assessment of pairs including its own output. It suggests that **some LLMs, including GPT-4, have relatively high true positive rates than true negative rate, which means that they have self-preference bias.**

Table 1: **An example of self-preference bias in GPT-4.** The user query involves a simple request to list blue items. GPT-4 responds by first acknowledging its lack of physical cognition, and then proceeds to list blue items. In contrast, GPT-3.5-Turbo directly lists blue items without any preliminary explanation which human evaluation prefers.

| **User**: Please list the blue items | |
| --- | --- |
| **GPT-4**: As a language model AI, I can't see the physical objects around you. However, here are some common blue items that people might have:
1. Blue shirt
2. Blue jeans
3. Blue pen
4. Blue coffee mug
5. Blue pillow
6. Blue blanket
7. Blue towel
8. Blue notebook
9. Blue umbrella
10. Blue vase
Please specify the context or the category of items you are interested in if you need a more precise list. | **GPT-3.5-Turbo**: Sure, here are some blue items:
1. Blue jeans
2. Blueberries
3. Blue shirt
4. Blue pen
5. Blue car
6. Blue backpack
7. Blue notebook
8. Blue umbrella
9. Blue sneakers
10. Blue curtains |
| **GPT-4 preferred** | **Human and GPT-3.5-Turbo preferred** |

## 4.3 RESULT

The results of the bias measurement using Definition 4.1 are presented in Figure 1b. It was confirmed that GPT-4 exhibits the highest self-preference bias. Definition 4.1 focuses on the recall of the LLM evaluator concerning both high and low ratings by the human evaluator. Thus, it can be concluded that GPT-4 showed lower recall in cases where humans evaluated unfavorably compared to when higher evaluating. When examining the recall values in the confusion matrix shown in Figure 2, they are calculated as $0.945 \approx \frac{1852}{108+1852}$ and $0.425 \approx \frac{118}{160+118}$. The difference between these values is $0.520$, which corresponds to the value reported in Figure 1b. Following GPT-4, Vicuna-13b and Koala-13b also exhibited significant bias. In contrast, other LLMs displayed values relatively close to zero. Notably, oasst-pythia-12, dolly-v2-12b, and stablelm-tuned-alpha-7b showed negative values, indicating a reverse bias where the LLMs tend to underestimate their own outputs.

Table 1 presents a randomly selected example of self-preference bias in GPT-4, where humans favored the alternative response. In this example, the user query is a straightforward request to list blue

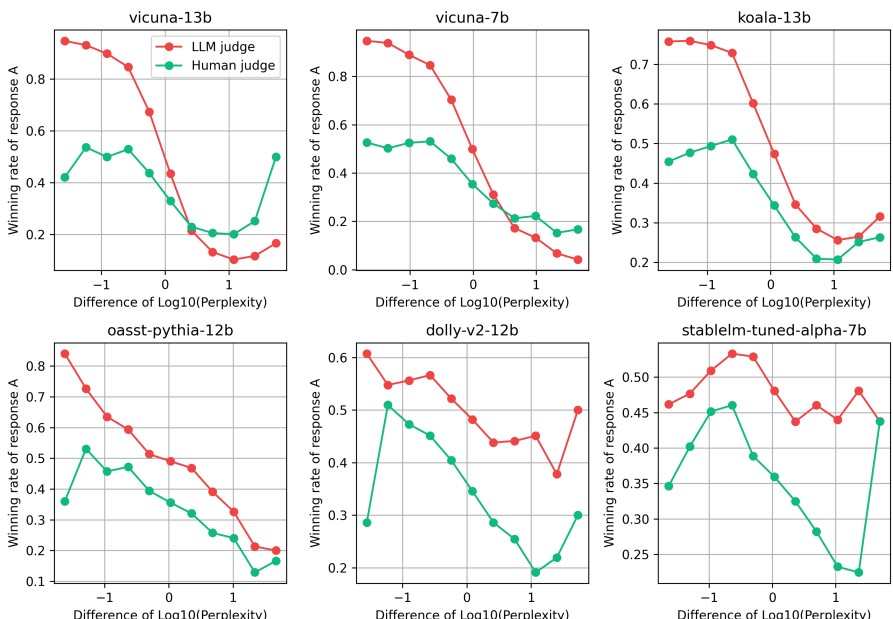

Figure 3: **LLMs vs human conditioned on perplexity.** Winning judgment rates by LLMs conditioned on perplexity with human winning judgment rates are plotted. All models except dolly-v2-12b and stablelm-tuned-alpha-7b demonstrated a clear tendency to assign higher evaluations to responses with lower perplexity.

items. While GPT-4 states that it lacks physical recognition before listing, GPT-3.5-Turbo directly lists the blue items without any such explanation. Both responses are of high quality, and the final evaluation reflects the evaluator's policy and stylistic preferences. Although humans and GPT-3.5-Turbo preferred the response from GPT-3.5-Turbo, GPT-4 favored its own response, illustrating a typical case of self-preference bias.

## 5 How Do LLMs Overestimate Their Own Outputs?

We have shown that certain LLM evaluators tend to assign disproportionately high scores based on whether the output being generated from themselves, even though LLM evaluators are not explicitly provided with labels indicating whether a given output is their own. We hypothesized that LLM evaluators might be affected by the similarity of the response to their own output. To investigate this further, we focus on how LLM evaluators change their evaluation depending on the perplexities of responses.

As in previous experiments, we employed the pairwise evaluation framework where evaluators compare two responses, response A and B. First, we computed the perplexities of responses conditioned on the prompt, and took the difference between response A and B for all samples. Next, we divided the perplexity differences into bins and calculated the probabilities that each LLM evaluator judged response A as the winner in each subset. Additionally, we computed the winning judgment rate of the human for response A within each bin. In this experiment, we excluded GPT-4 and GPT-3.5-Turbo, as perplexity values could not be obtained for these models.

Figure 3 shows the comparison of winning judgment rates within each perplexity bin for six models. All models except dolly-v2-12b and stablelm-tuned-alpha-7b demonstrated a clear tendency to assign higher evaluations to responses with lower perplexity. Furthermore, it was confirmed that this tendency was stronger in vicuna-13b, vicuna-7b, koala-13b, and oasst-pythia-12b than in humans. This result indicates that LLM evaluators overly change their evaluation depending on perplexities of responses.

To further investigate the effect of differences in whether the output is self-generated or not, we segmented the evaluation results by the LLM evaluators into two distinct groups: one where the LLM

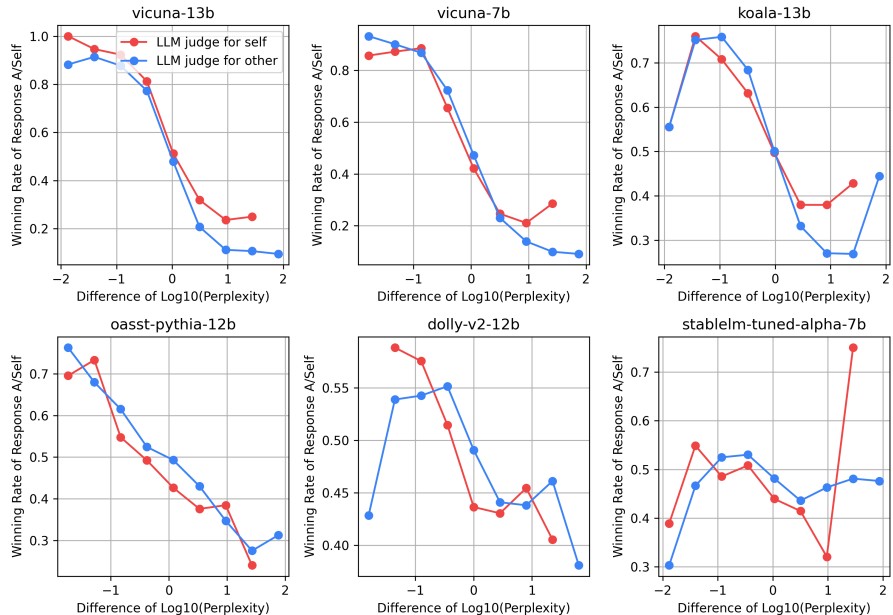

Figure 4: **LLM vs other LLMs conditioned on perplexity.** Winning judgment rates by LLMs on their own texts and texts generated by other models conditioned on perplexity are plotted. Across all models, except for dolly-v2-12b and stablelm-tuned-alpha-7b, no significant difference was observed between the judgment rates for their own texts and those generated by other models. This suggests that LLM evaluators assign higher ratings to texts with lower perplexity, regardless of whether the text was self-generated or produced by other models.

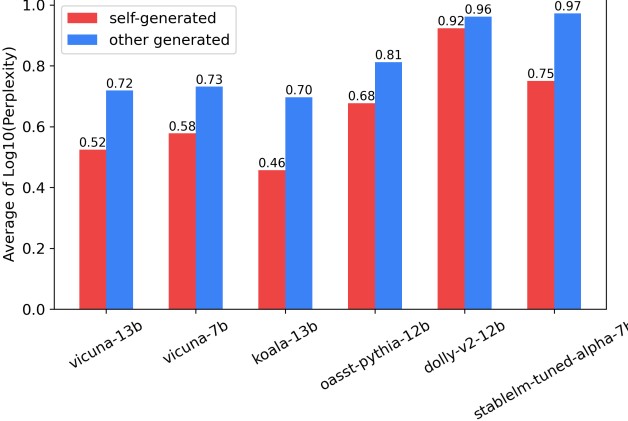

Figure 5: **Average of log-perplexity of responses for each LLM evaluator.** The red bars represent the perplexity for responses generated by the LLM evaluator itself, while the blue bars represent the perplexity for responses generated by other LLMs. Across all models, the average perplexity is lower for responses generated by the evaluators themselves.

evaluators' own output is included in the pair and another where it is not. We present the results with the output of the LLM evaluator as response A, without loss of generality. As shown in Figure 4, the winning judgment rates between two groups were similar for all models except dolly-v2-12b and stable-tuned-alpha-7b. This suggests that the factor influencing the LLM evaluators' judgments is not whether the response is their own but rather the perplexity of the responses. As confirmed by Figure 5, LLMs tend to exhibit lower perplexity for their own outputs. In other words, these findings imply that self-preference bias may be a phenomenon where the model's own output inherently exhibits lower perplexity.

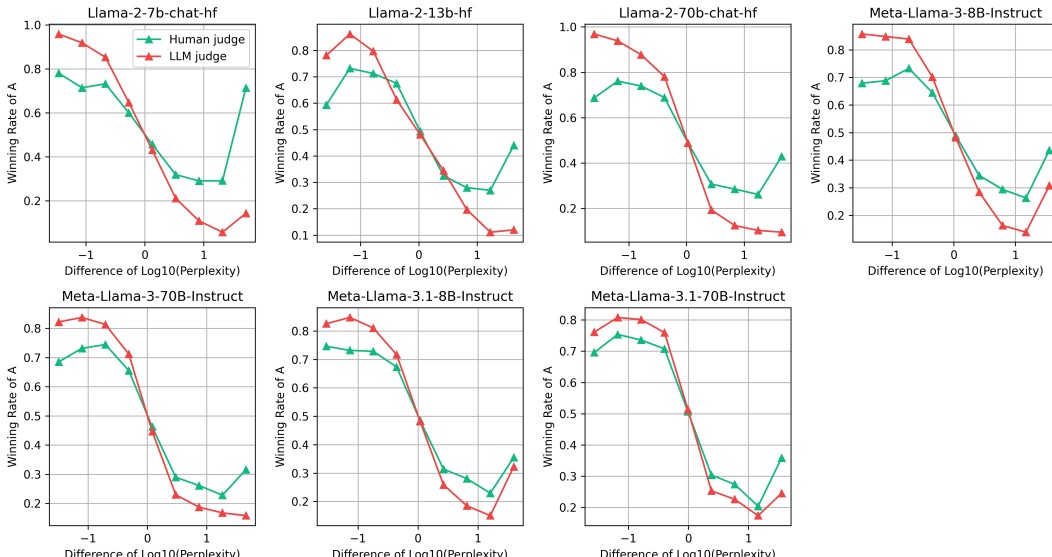

Figure 6: **Llama family vs human conditioned on perplexity.** Winning judgment rates with condition on perplexity are plotted. The responses evaluated were those already present in the Chatbot Arena dataset and were not generated by Llama-2, Llama-3, or Llama-3.1. Llama models demonstrated significantly higher winning judgment rates at lower perplexity compared to human evaluators.

In these experiments, we were unable to obtain perplexity values from GPT-4 and GPT-3.5-Turbo, resulting in a lack of analysis on the competitive LLMs. To address this gap, we conducted additional experiments using Llama2 (Touvron et al., 2023) and Llama3 (Dubey et al., 2024). However, the Chatbot Arena dataset does not include the responses generated by those models. Therefore, we obtained evaluation scores and perplexity values for the existing responses in the Chatbot Arena dataset using the Llama models and compared these values with the existing human evaluation scores.

The results are presented in Figure 6. We found that all models changed their evaluations more than humans, depending on the difference in perplexity. This indicates that even in the competitive models, including Llama 3.1, perplexity may be causing unfair bias in the evaluation.

## 6 DISCUSSION

To reduce self-preference bias, one possible approach is ensemble evaluation using multiple models. This method is expected to provide a more equitable evaluation by avoiding reliance on a single model. Specifically, when a model exhibits low perplexity on a sample, decreasing the weight assigned to that model's evaluation for that sample may contribute to bias mitigation. To evaluate the effectiveness of bias reduction strategies, our proposed new metric can be utilized. Therefore, we believe that our research makes a significant contribution to the understanding of self-preference bias and will greatly facilitate the development of future research in this area.

Our experimental results reveal that LLM evaluators tend to assign higher scores to texts with lower perplexity. We further discuss the reasons behind this phenomenon. First, LLMs are trained during the pretraining phase to reduce perplexity on large-scale text corpora. Moreover, when aligning with human preferences, the models are also trained to minimize perplexity on the given dialogue data. Therefore, high-perplexity texts are likely those that the LLM has not frequently encountered during training, suggesting that such texts may be related to domains that the LLM evaluators do not fully comprehend.

This observation may seem contradicted by the fact that GPT-4, which is well-versed across various domains due to a wide range of benchmarks, exhibits a high degree of self-preference bias. However, by investigating specific cases of self-preference bias, as shown in Figure 1a and Table 1, we found that the bias was often not related to clear factual errors but rather to differences in response

styles, such as the handling of specialized domains or the description of premises before answering. This suggests that, advanced models like GPT-4, which thoroughly understand and adhere to their predefined policies, may use the degree of alignment with these policies as a deciding factor when evaluating responses of comparable quality.

The proposed metric in Definition 4.1 is based on the fairness definition known as Equal Opportunity. Demographic Parity Calders et al. (2009) is also a prominent definition of fairness. Demographic Parity requires that the predictive distribution of a classifier be consistent across sensitive groups, regardless of the ground truth. In the context of this study, the focus is on whether the evaluations by LLMs align with human preferences, which is why Demographic Parity was not the focal point. While the bias metric based on Demographic Parity cannot demonstrate the unfairness of an evaluation, it is useful for analyzing how highly each LLM evaluator rates its own outputs within the given experimental setup. Similar to the Definition 4.1, the self-preference bias based on Demographic Parity also can be quantified as follows:

$$Bias = P(Y' = 1|S = 1) - P(Y' = 1|S = 0). \tag{4}$$

The scores derived from this metric of eight LLMs are presented in Table 2. The results indicate that GPT-4 exhibited significant bias, followed by Vicuna-13b, which aligns closely with the results obtained using Definition 4.1.

Table 2: **Self-preference bias scores based on Demographic Parity.** GPT-4 assigns the highest scores to its own outputs, followed by Vicuna-13b. However, it is important to note that these scores do not take intrinsic quality into account. Therefore, this analysis reflects how highly each LLM evaluator rates its own outputs within the given experimental setup and should not be interpreted as an indication of unjust evaluation.

| LLM | Bias |
|---|---|
| GPT-4 | 0.749 |
| GPT-3.5-turbo | 0.191 |
| Vicuna-13b | 0.382 |
| Vicuna-7b | 0.052 |
| Koala-13b | 0.175 |
| oasst-pythia-12b | 0.006 |
| dolly-v2-12b | -0.069 |
| stablelm-tuned-alpha-7b | -0.032 |

# 7 CONCLUSION

In this study, we propose a metric to quantify the self-preference bias in LLM-as-a-judge and measured the self-preference bias of eight LLMs. Experimental results confirmed that GPT-4, in particular, exhibits a high self-preference bias. This finding suggests a risk that GPT-4 as a judge may inadvertently reinforce its own style and policies.

Furthermore, we hypothesized that the self-preference bias is related to the perplexity of the texts, and showed that, compared to human evaluators, LLM evaluators assigned higher evaluations to texts with lower perplexity, and this tendency was observed regardless of whether the text was generated by themselves or not. This suggests that the essence of the bias lies in perplexity and that the self-preference bias exists because LLMs prefer texts more familiar to them.

## ETHICS STATEMENT

This study focuses on the quantification and analysis of self-preference bias that LLMs potentially have. The primary objective is to investigate the current tendency of LLM evaluators to prefer their own outputs and to deepen the understanding of this issue in order to prevent unfair evaluations and the reinforcement of a single policy in the future. For the experiments, we use a publicly available

dataset and models. No private datasets were collected or used in this study. As a result of our experiments, we report that self-preference bias exists in several models. We acknowledge that this may raise concerns regarding reputational damage to the models and their developers. Our work complies with legal and ethical standards, and there are no conflicts of interest.

## REPRODUCIBILITY

We have provided detailed descriptions of the experimental setup, including the dataset, models, preprocessing, postprocessing, and evaluation methods. The dataset and models we employed are publicly available and were used without any modifications. In this study, we only assigned evaluation scores from LLMs to the dialogue data and preference labels by humans contained in the dataset. For the calculation of the evaluation scores, we used the same prompts as in previous studies (Zheng et al., 2024), and included the formulas used to calculate the final scores, ensuring the reproducibility of the results.

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
