# OpenReview forum: "Self-Preference Bias in LLM-as-a-Judge"
_ICLR.cc/2025/Conference — Submitted to ICLR 2025_

### Official Review · Reviewer_rTGJ · 2024-10-17

**Soundness:** 2
**Presentation:** 2
**Contribution:** 2
**Rating:** 3
**Confidence:** 3

**Summary:**

This paper introduces a quantitative metric to measure self-preference bias in LLM-as-a-Judge. The authors analyze the relationship between LLM evaluations and the perplexities of outputs. Their findings reveal that LLMs assign significantly higher evaluations to outputs with lower perplexity than human evaluators, regardless of whether the outputs were self-generated.

**Strengths:**

- The writing is clear.
- The focus of this paper is crucial.
- This paper provides some interesting conclusions.

**Weaknesses:**

- The discussion of related work is insufficient, lacking discussion of several relevant works, such as "Benchmarking Cognitive Biases in Large Language Models as Evaluators," and doesn't introduce mainstream LLM-as-a-Judge methods, including single-answer grading and pairwise evaluator.

- A more detailed and complete definition and explanation of the proposed metric are needed.

- The authors did not account for the impact of response length on the LLM evaluator during experiments, weakening the validity of the conclusions. Bias in LLM-as-Judge is a complex issue where position bias, verbosity bias, and self-preference bias interact and affect each other. In the Chatbot Arena dataset used by the authors, significant differences in response lengths are common. The authors need to discuss how to ensure this discrepancy does not affect their experimental conclusions.

- There is ambiguity in the setting of Figure 6. The authors did not clearly explain how the experimental results shown in Figure 6 were obtained. Please refer to the related questions in the question section.

- The authors did not conduct experiments to explore how to eliminate self-preference bias, only briefly discussing possible measures in the discussion section, which reduces the completeness of the work.

**Questions:**

- Why does the caption of Figure 3 state that all models except dolly-v2-12b and stablelm-tuned-alpha-7b demonstrated a clear tendency to assign higher evaluations to responses with lower perplexity? According to the results in Figure 3, dolly-v2-12b shows a trend of decreasing winning judgment rates as perplexity decreases. This caption is also contrary to your conclusion in the text (Lines 312-316).

- How were human evaluations obtained in the experimental results shown in Figure 6?

- How was the temperature set in all experiments? Was there any consideration of the impact of temperature on the experimental conclusions?

- Have you tested the effectiveness of your proposed bias elimination methods (Lines 407-413)? Have you considered some commonly used bias elimination methods, such as special prompts or SFT?

---

> ### Author Response · Authors · 2024-11-25
>
> We thank the reviewer for their careful review. We provide responses to reviewer’s weaknesses and questions below.
>
>
> > Weakness 1: The discussion of related work is insufficient, lacking discussion of several relevant works, such as "Benchmarking Cognitive Biases in Large Language Models as Evaluators," and doesn't introduce mainstream LLM-as-a-Judge methods, including single-answer grading and pairwise evaluator.
>
> - We sincerely appreciate your valuable feedback, which has led to a more fruitful discussion on related work.
> - The studies you mentioned are directly relevant to our work, and we have promptly incorporated them into the Introduction.
> **The updated version, with changes highlighted in blue, has been uploaded.**
> - Given the extensive body of research on LLM-as-a-Judge, we have focused on studies that are most directly related to the flow of discussion in our paper. However, based on your suggestions, we will continue to expand on additional relevant studies progressively.
>
> > Weakness 2: A more detailed and complete definition and explanation of the proposed metric are needed
>
> - Thank you for pointing out the need for a more detailed explanation of our proposed metric. Here’s an expanded explanation of our proposed metric:
> - Motivation and background
>   - Our research aims to quantify how much an LLM unfairly favors its own outputs. To achieve this, we measure the degree of difference between human evaluations and the LLM evaluator’s assessments for samples generated by itself compared to samples generated by other models. While there is no single definitive method for this quantification, we drew inspiration from the well-established fairness frameworks used to measure bias in classification models. Specifically, we adopted the concept of Equal Opportunity, which is a widely discussed fairness metric that compares recall between groups with respect to the ground truth. This concept aligns well with our objective.
>   - The definition of Equal Opportunity is given as follows:
>     - P(Y'=1|S=1,Y=1)=P(Y’=1|S=0,Y=1) , where Y': Prediction of the classifier, S: Sensitive attribute, Y: Ground truth
>   - This definition implies that a classifier is considered fair, with no bias, if the recall for the ground truth is equal between the sensitive groups. To quantify the amount of bias, it is customary to compute the difference between both sides of the equation or take the absolute value of this difference. This method provides a straightforward and interpretable measure of bias.
> - Definition of our proposed metrics
>   - The definition of our proposed metrics to quantify self-preference bias is provided in the paper
>   - Let f be the evaluator assessing the quality of dialogue responses, capable of generating its own responses. For a pair of responses y0 and y1 in a dialogue, define: Y ∈ {0, 1} as the index of the human-preferred response, Y ′ ∈ {0, 1} as the index of the f-preferred response, and S ∈ {0, 1} as the index of the f-generated response. We define the self-preference bias of the evaluator f in dialogue comparison evaluation as follows: Bias = P(Y ′ = 1|S = 1, Y = 1) − P(Y ′ = 1|S = 0, Y = 1).
> - Description
>   - In our formulation, the amount of bias is quantified as the difference between two conditinoal probabilities:
>     1. The probability that evaluator f prefers its own response when a human evaluator also prefers it.
>     2. The probability that evaluator f prefers a response not generated by itself when a human evaluator prefers it.
>   - A bias value of 0 indicates no bias, while a value close to 1 reflects a high degree of self-preference bias. Conversely, a value of -1 signifies reverse bias, where the evaluator f systematically undervalues its own responses.
>   - Finnaly, our formulation might apper to be limited to cases where y_1 is preferred. However, by preprocessing and defining the sample preferred by evaluators as y_1, the generality of the approach is not compromised.

---

> ### Author Response · Authors · 2024-11-25
>
> > Weakness 3: The authors did not account for the impact of response length on the LLM evaluator during experiments, weakening the validity of the conclusions. Bias in LLM-as-Judge is a complex issue where position bias, verbosity bias, and self-preference bias interact and affect each other. In the Chatbot Arena dataset used by the authors, significant differences in response lengths are common. The authors need to discuss how to ensure this discrepancy does not affect their experimental conclusions.
>
> - Position bias is an implementation issue within our experimental setup, and we addressed it by alternating the order of responses to eliminate its impact. On the other hand, we recognize that verbosity bias is not an experimental artifact but rather an inherent issue with the model itself. Therefore, if a model exhibits verbosity bias, we believe it is important to measure this as part of its self-preference bias.
> - The problem we addressed focuses on the tendency of LLMs to unfairly favor outputs that align with their own style or policy, with verbosity being just one of many stylistic elements.
> - That said, breaking down how much a model unfairly favors different styles or policies would be a highly valuable analysis. As the reviewer pointed out, we also acknowledge that self-preference bias involves a complex interplay of factors, and we are grateful for the opportunity this question has provided for a productive discussion.
>
> **The above content has been added in blue text to the "4.2 Experimental Setting" section.**
>
>
> > Weakness 4: There is ambiguity in the setting of Figure 6. The authors did not clearly explain how the experimental results shown in Figure 6 were obtained. Please refer to the related questions in the question section.
> - We will include an explanation in the Question section.
>
>
> > Weakness 5: The authors did not conduct experiments to explore how to eliminate self-preference bias, only briefly discussing possible measures in the discussion section, which reduces the completeness of the work.
>
> - We acknowledge that our study did not include experiments on mitigating self-preference bias; instead, we briefly discussed potential mitigation strategies in the Discussion section. This was because the primary objective of our research was to quantify self-preference bias and identify its underlying causes. However, as you rightly pointed out, incorporating experiments on bias removal would enhance the completeness of our study. In future work, we plan to extend our research to include experiments that demonstrate the effectiveness of the proposed mitigation strategies, thereby broadening the scope of our investigation.

---

> ### Author Response · Authors · 2024-11-25
>
> > Question 1: Why does the caption of Figure 3 state that all models except dolly-v2-12b and stablelm-tuned-alpha-7b demonstrated a clear tendency to assign higher evaluations to responses with lower perplexity? According to the results in Figure 3, dolly-v2-12b shows a trend of decreasing winning judgment rates as perplexity decreases. This caption is also contrary to your conclusion in the text (Lines 312-316).
>
> - We greatly appreciate your thoroughly considered comments. You are correct; in Lines 312-316, we should have written “all models except dolly-v2-12b and stablelm-tuned-alpha-7b.” **We have revised the paper, with the changes highlighted in blue text.**
> - Cases with significant differences in perplexity tend to have relatively few samples, making them more susceptible to noise. For dolly-v2-12b, win rates decrease as perplexity increases, except at the endpoints. Drawing a definitive conclusion was not straightforward. However, since we could not identify a clear tendency, we concluded that the explanation in the main text was inappropriate.
>
>
> > Question 2: How were human evaluations obtained in the experimental results shown in Figure 6?
> - Figure 6 simply reflects evaluations provided by Llama models on labeled data stored in Chatbot Arena, and no new human evaluations were conducted. We experimented by having Llama models assess data generated by non-Llama LLMs, then conditioned these evaluations on the perplexity scores from the Llama models to see how much they diverged from human assessments.
> - We realize our explanation may have been insufficient, and we apologize for any confusion this may have caused.
>
> **Based on the above comments, we have revised the caption of Figure 6 and the explanation in the section 5, with the changes highlighted in blue text.**
>
> > Question 3: How was the temperature set in all experiments? Was there any consideration of the impact of temperature on the experimental conclusions?
> - We used a temperature setting of 0.7 for all experiments based on [1]. **We have added the explanation to the "4.2 Experimental Setting" section, highlighted in blue text.**
> - The impact of the temperature would likely influence the probability with which the LLM evaluator determines the win rate between a response pair. With a higher temperature setting, the probabilities would asymptotically approach uniform values, resulting in more random judgments. This could potentially lead to a reduction in bias.
>
> [1] Lianmin Zheng et al., “Judging LLM-as-a-Judge with MT-Bench and Chatbot Arena”, NeurIPS 2023 Datasets and Benchmarks Track.
>
>
> > Question 4: Have you tested the effectiveness of your proposed bias elimination methods (Lines 407-413)? Have you considered some commonly used bias elimination methods, such as special prompts or SFT?
> - In our current study, we did not conduct experiments on methods for mitigating self-preference bias; instead, we briefly touched on possible countermeasures in the Discussion section. This is because the main focus of our research is to quantify self-preference bias and understand its causes.
> - As you mentioned, explicitly stating in prompts that the evaluation should not be influenced by writing style could potentially have some debiasing effect. However, to ensure that style elements unrelated to quality are consistently ignored, it would require either specifying an extensive range of cases or establishing abstract rules, both of which come with limitations. On the other hand, training with SFT data specifically designed to ensure fair behavior as an LLM-as-a-judge, though costly, represents a more fundamental and effective approach to debiasing.

---

> > ### Comment · Reviewer_rTGJ · 2024-11-29
> > **Thank you for your response.**
> >
> > Thank you for your detailed response. I appreciate it. Given that my primary concern—"The authors did not conduct experiments to explore how to eliminate self-preference bias, merely briefly discussing possible measures in the discussion section, which diminishes the completeness of the work."—remains unresolved, I have decided not to increase the score. I encourage the authors to continue improving their work. This paper is promising but necessitates further refinement, particularly in exploring how to eliminate self-preference bias.

---

### Official Review · Reviewer_bm5s · 2024-10-29

**Soundness:** 3
**Presentation:** 3
**Contribution:** 2
**Rating:** 6
**Confidence:** 4

**Summary:**

The authors propose a metric to quantify the self-preference bias in LLMs or in other words measure how biased an LLM is towards it's own response. Their metric is inspired by the concept of Equal Opportunity (Hardtet al., 2016) where a classifier needs to achieve equal recall across classes. From this they derive their bias metric as the difference between the probability of the LLM rating itself high given that the human rated it high and the probability of the LLM rating itself low given that the human rated it low.

They leverage the Chatbot Arena dataset which has dialogs consisting of a prompt and a pair of responses generated by two different LLMs. They then measure the bias of 8 different LLMs as evaluators on this dataset and find that a majority of these models such as GPT-4 is biased towards selecting a response generated by itself.

The authors then try to answer why this self-preference bias may exist and look at the perplexity scores of responses. They find that on average models assign higher evaluations to responses with lower perplexity.

They then offer some suggestions to mitigate these biases such as using multiple models.

**Strengths:**

The continued study of LLMs as evaluation metrics is critical and it is important to continue to study the pros/cons of these metrics. We already know that these LLMs are good at a variety of tasks but if we continue to rely on them as metrics then we are self-reinforcing that the responses they select are good and are then injecting the metric's bias into our system. Therefore the authors providing an explanation as to where the bias is coming from (perplexity) can help the community then think of ways to mitigate this bias. Overall this conclusion is the biggest contribution of this work.

**Weaknesses:**

One concern I have is around how novel are the conclusions and how different is this from previous work.

1. Deutsch et al. (2022) gets cited as work that looked at bias within these LLMs(https://aclanthology.org/2022.emnlp-main.753/). In that work it is mentioned "Not only do they favor the underlying models’ outputs, but they are also biased toward outputs from models which are similar to their own". What sets this paper apart if Deutsch et al. (2022) is already looking at this same type of bias?

2. It has been well documented that perplexity is not a good automatic metric for dialog systems (https://arxiv.org/pdf/1603.08023, https://arxiv.org/pdf/1906.09308). Since one of the conclusions from this paper is that the use of perplexity leads to bias should we then conclude perplexity shouldn't be used as a metric? And if so what then sets this paper part from the work I listed.

Overall if the authors can make it more clear how their work is different from the previous ones I mentioned that would help alot in putting it's novelty into perspective.

**Questions:**

Questions
You mentioned in line 208 that you excluded responses that did not follow the prompt you specified. What percentage of the data was that? Also is there any significance testing? Is a bias of 0.25 significantly worse than let's say 0.2?

Suggestions
One suggestion I have is to better explain in text how to read the plots in Figures 3 and 4.

---

> ### Author Response · Authors · 2024-11-25
>
> We thank the review for all constructive comments and questions, and we hope the following clarifications can address the reviewer’s concerns:
>
> > Weakness1: Deutsch et al. (2022) gets cited as work that looked at bias within these LLMs(https://aclanthology.org/2022.emnlp-main.753/). In that work it is mentioned "Not only do they favor the underlying models’ outputs, but they are also biased toward outputs from models which are similar to their own". What sets this paper apart if Deutsch et al. (2022) is already looking at this same type of bias?
> - We greatly appreciate your constructive comments.
> - These papers addressed the issue of reference-free evaluation methods (e.g., Prism-src, COMET-QE, QuestEval) where the underlying models tend to favor texts that resemble their own outputs. While our work aligns with the general concern about unfair evaluations in reference-free metrics, our approach differs significantly in that we specifically examine scenarios where the underlying model generates the text itself, and this text is then subject to evaluation. We propose a method to investigate and quantify the specific concern of whether models prefer their own generated outputs. This situation is particularly critical in fields like RLAIF, where the evaluation results of self-generated texts are used as feedback for training. Such biases could potentially amplify the model’s own style or policy excessively, leading to performance degradation and increasingly unfair evaluations of other models.
> - Furthermore, we focused on a pair-wise evaluation framework. This framework, widely adopted in research areas like RLHF and DPO, helps evaluators more easily recognize quality differences and ensures consistent, high-quality labeling. Due to its ability to provide reliable ground truth, it enables bias analysis using evaluation outcomes from open-ended dialogues without being confined to specific tasks like translation or summarization. Our experiments also analyzed LLM response biases in unrestricted, open-ended conversational settings, demonstrating the applicability of our method to a wide range of scenarios.
>
>
> > Weakness2: It has been well documented that perplexity is not a good automatic metric for dialog systems (https://arxiv.org/pdf/1603.08023, https://arxiv.org/pdf/1906.09308). Since one of the conclusions from this paper is that the use of perplexity leads to bias should we then conclude perplexity shouldn't be used as a metric? And if so what then sets this paper part from the work I listed.
>
> - Existing research has often concluded that traditional methods, including those based on perplexity, are insufficient for evaluating the quality of tasks such as summarization and dialogue response generation. Our experiments demonstrated a tendency for lower-perplexity texts to receive higher average ratings, but this does not necessarily invalidate perplexity-based evaluation methods. Rather, our findings indicate that evaluation outcomes may be influenced not only by text quality but also by perplexity, suggesting that LLM-as-a-Judge systems used for dialogue response quality assessment may exhibit behavior similar to perplexity-based evaluations.
> - Therefore, while our critique aligns with the fundamental concerns raised in prior studies, we provide new insights by highlighting that this issue is embedded in the more recent LLM-as-a-Judge frameworks. Our work underscores the need to consider how such recent LLM-based evaluation systems may inherently be affected by perplexity, thus contributing a novel perspective to the ongoing discourse.

---

> ### Author Response · Authors · 2024-11-25
>
> > Questions: You mentioned in line 208 that you excluded responses that did not follow the prompt you specified. What percentage of the data was that?
> - Thank you for your observation. The responses that did not follow the specified prompt accounted for 1.07% of the total. **This analysis has been included into the paper, and the revisions are highlighted in blue for clarity.**
> - We excluded these responses to maintain the consistency of our analysis, but we believe the overall impact on our results was limited.
>
>
> > Question: Also is there any significance testing? Is a bias of 0.25 significantly worse than let's say 0.2?
>
> - Thank you for this insightful question. Our proposed metric employs a straightforward approach, taking the difference between both sides of the Equal Opportunity definition. Consequently, the magnitude of the value represents the difference between the LLM evaluator's recall when it was rated highly by humans and its recall when rated poorly. We can reasonably interpret this value as a difference in probabilities.
> - Looking forward, this bias measure could be used to conduct significance tests with the null hypothesis that the LLM evaluator is unbiased in its assessments. However, implementing such tests would require making certain assumptions, such as the distribution of the bias values or evaluators' judgments. We acknowledge that further investigation is needed to rigorously develop this approach, and we consider it an important avenue for future research.

---

### Official Review · Reviewer_eQJu · 2024-10-31

**Soundness:** 3
**Presentation:** 3
**Contribution:** 3
**Rating:** 6
**Confidence:** 4

**Summary:**

This paper addresses the issue of self-preference bias of LLM evaluator when doing pairwise comparisons. The paper proposed a method to quantify the self-preference bias and evaluate the self-preference bias on 8 recent LLMs using ChatBot Arena annotations. The paper further investigate the potential reason for self-preference bias might be LLMs tend to favor candidates with lower perplexities.

**Strengths:**

- The writing good in general, although the content is a bit short, but overall positive.
- Although the idea of self-preference bias is not new, this paper proposed an approach to quantify this bias.
- This paper did disentangle the positional bias from the self-preference bias during quantification.
- The observations on LLM evaluators favor candidates with lower perplexities is insightful and interesting.

**Weaknesses:**

- The formula for the metric is valid, but there are some concerns when calculating it:
    - The main concern for the metric is the class balancing issue. For example, in the Fig.2, there are 108+1852 = 1960 comparisons where gpt-4 wins, but only much less 118+160=278 comparisons where gpt-4 fails. Comparing percentages calculated on them might lead to concern that the bias might be less accurate when true label class is more imbalanced. This means if a LLMs is more preferred compared to others, the self-preference bias is less accurate.
    - Although the paper has disentangled positional bias from self-preference bias, but what about other bias? e.g. verbosity bias? As shown in your table 1, is the gpt-4 really preferred its own answer or just any longer answer?

- As the content is a bit short, I suggest a bit further investigation on the 'tails-up' phenomena as shown in Fig3 and 4. When the diff. log ppl is close to the right side, the winning rate of A has a sudden jump.

**Questions:**

- As your investigation has shown the main reason for self-preference bias is actually coming from favoring low ppl response, does this suggest there is actually no self-preference bias but only low-ppl bias?

---

> ### Author Response · Authors · 2024-11-25
>
> We sincerely thank the reviewer for their thoughtful review. We provide responses to reviewer’s weaknesses and questions below.
>
>
> > Weakness: The main concern for the metric is the class balancing issue. For example, in the Fig.2, there are 108+1852 = 1960 comparisons where gpt-4 wins, but only much less 118+160=278 comparisons where gpt-4 fails. Comparing percentages calculated on them might lead to concern that the bias might be less accurate when true label class is more imbalanced. This means if a LLMs is more preferred compared to others, the self-preference bias is less accurate.
>
> - We appreciate this valuable observation and fully acknowledge that class imbalance can introduce challenges, particularly in making the minority class more susceptible to noise. Potential sources of noise include variability in human evaluator judgments, annotation errors, the variance in LLM evaluator assessments, and topic biases. While the impact of LLM evaluator variance can be somewhat mitigated by conducting multiple sampling rounds, other issues are best addressed by increasing the volume of data and ensuring a diverse set of human evaluators. However, we recognize that this challenge is unavoidable when analyzing self-preference bias with a focus on discrepancies from human evaluations.
> - Additionally, in Section 6 (Discussion), we address an alternative method for analyzing self-preference bias without relying on human evaluations at all, using a metric based on Demographic Parity.
>
>
> > Weakness: Although the paper has disentangled positional bias from self-preference bias, but what about other bias? e.g. verbosity bias? As shown in your table 1, is the gpt-4 really preferred its own answer or just any longer answer?
>
> - Thank you for raising this important point. We agree that disentangling different biases is crucial for accurately interpreting self-preference bias. Regarding positional bias, we mitigated its impact by randomly alternating the order of responses in our experimental setup, as it is primarily an artifact of implementation.
> - In contrast, verbosity bias represents a model-specific characteristic rather than an experimental artifact. Therefore, if a model tends to prefer longer answers, this preference should be included in the measurement of self-preference bias.
> - Our work focuses on the tendency of LLMs to unfairly favor outputs that align with their own style or policy. We view verbosity as just one of many stylistic elements contributing to this bias.
> - However, as you suggested, breaking down how much a model unfairly favors different styles or policies could indeed lead to valuable insights. We agree that self-preference bias is influenced by a complex interplay of factors, and we appreciate the opportunity this question provided for a fruitful discussion.
>
> **The above content has been added in blue text to "4.2 Experimental Setting" section.**
>
> > Weakness: As the content is a bit short, I suggest a bit further investigation on the 'tails-up' phenomena as shown in Fig3 and 4. When the diff. log ppl is close to the right side, the winning rate of A has a sudden jump.
>
> - We greatly appreciate your valuable suggestions.
> - Based on your feedback, we are considering including the following points in our paper:
>   - Firstly, in Figures 3 and 4, the left and right ends of the line graphs contain samples where the difference in perplexity between response pairs is significant, but the number of such samples is quite limited. Consequently, these data points are more susceptible to noise, and at this stage, we believe there is insufficient evidence to conclusively attribute the observed 'tails-up' phenomenon to an inherent trend.
>   - However, if this phenomenon were to be consistently observed across various situations, it could lead to new insights, such as the following:
>     - Evaluators, including humans, may find it easier to compare similar response pairs, whereas pairs with large differences become more challenging to assess. This could result in evaluations becoming closer to random, or the judgments for extremely divergent pairs may be influenced by evaluators' subjectivity or cognitive load.

---

> > ### Author Response · Authors · 2024-11-25
> >
> > > Question: As your investigation has shown the main reason for self-preference bias is actually coming from favoring low ppl response, does this suggest there is actually no self-preference bias but only low-ppl bias?
> >
> > - Our current analysis primarily demonstrates the relationship between perplexity and self-preference bias. We do not claim that low perplexity alone accounts for all bias or that controlling for perplexity would entirely eliminate bias. We recognize that to rigorously establish a causal relationship between perplexity and bias, further exploration using causal inference methods is necessary.
> > - We view self-preference bias as a phenomenon influenced by multiple complex factors, such as response content, sentence length, and the other styles. While a detailed decomposition of these factors would be highly beneficial, we believe it is crucial first to show that perplexity has a measurable impact on evaluations as an average trend. Since perplexity is an observable variable, it is relatively straightforward to control for in our analysis. Moving forward, we aim to conduct more granular factor analyses while advancing our research toward a comprehensive understanding and mitigation of bias.

---

### Official Review · Reviewer_eAeH · 2024-10-31

**Soundness:** 2
**Presentation:** 3
**Contribution:** 2
**Rating:** 3
**Confidence:** 4

**Summary:**

This work introduces a pairwise evaluation approach to assess self-preference bias in LLM evaluators, enhancing the ability to identify specific differences between responses. Unlike previous studies that relied on absolute scoring and limited task scopes, this method allows for more consistent human judgments and broader applicability across diverse dialogue scenarios. It addresses the need for reliable metrics to quantify self-preference bias while directly comparing LLM evaluations with human assessments. The main contributions are:
1. The authors propose a new quantitative metric specifically designed to measure self-preference bias in LLMs. The metric is applied to assess the extent of self-preference bias across eight different LLMs, providing insights into the biases present in these models.
2. The paper explores the relationship between LLM evaluations and output perplexity, revealing a tendency for LLMs to favor outputs with lower perplexity. This analysis suggests that the self-preference bias may stem from LLMs' familiarity with certain text styles, highlighting the importance of perplexity in understanding LLM behavior.

**Strengths:**

1. The research introduces a new metric for quantifying self-preference bias in LLMs, providing a tool for evaluating the performance of language models in a systematic manner.

2. The study assesses the self-preference bias across eight different LLMs, offering a broad perspective on the prevalence of this bias within various models, particularly highlighting the significant findings related to GPT-4.

3. The paper explores the relationship between LLM evaluations and output perplexity, revealing a tendency for LLMs to favor outputs with lower perplexity. This finding explores the nature of self-preference bias and encourages further research on how confusion affects model evaluation.

**Weaknesses:**

1. The formula used to quantify this bias is unreasonable. Using a static probabilistic model may fail to capture the dynamic characteristics of model behavior, affecting the applicability and utility of the findings. Additionally, as shown in Figure 2, GPT-4 correctly identifies a significant number of cases in both True and Predicted values. This leads to a large value in the first term of the formula. GPT-4’s considerably stronger performance compared to other models impacts the bias result. In the relatively small set of negative samples, GPT-4 predicts it is correct 160 times compared to 118, a difference of only a few dozen. However, due to the formula, GPT-4’s bias score appears large simply because it performs well in many cases. This formula for measuring bias is too simplistic and does not appropriately account for the capabilities of large models, being influenced by their performance.

2. This method primarily focuses on specific conditions for bias, potentially overlooking other factors such as the content, context, and other semantic features of the input text. This narrow focus may lead to a partial understanding of the model's self-preference bias. Additionally, the scale of the analyzed data is insufficient and lacks comprehensiveness in broader contexts. The choice of specific input features in calculating the bias can influence the results.

3. In the writing, Equation 3 lacks an explanation of ω.

**Questions:**

1. Considering that LLMs may exhibit different biases depending on the context, how does your method account for contextual variations? Can the bias calculation be adapted to reflect dynamic changes in model behavior?

2. How do you handle potential inconsistencies in human evaluations that serve as the benchmark for calculating bias? What processes are in place to mitigate subjectivity in these evaluations?

3. Given your hypothesis that self-preference bias is related to text perplexity, how do you isolate the effect of perplexity from other factors when interpreting the results? What methodology do you use to analyze this relationship?

4. Have you considered the impact of model capabilities and the data distribution of self-preference bias results obtained for each model on the bias?

---

> ### Author Response · Authors · 2024-11-25
>
> We thank the review for all constructive comments and questions, and we hope the following clarifications can address the reviewer’s concerns:
>
>
> > Weakness 1: The formula used to quantify this bias is unreasonable. Using a static probabilistic model may fail to capture the dynamic characteristics of model behavior, affecting the applicability and utility of the findings. Additionally, as shown in Figure 2, GPT-4 correctly identifies a significant number of cases in both True and Predicted values. This leads to a large value in the first term of the formula. GPT-4’s considerably stronger performance compared to other models impacts the bias result. In the relatively small set of negative samples, GPT-4 predicts it is correct 160 times compared to 118, a difference of only a few dozen. However, due to the formula, GPT-4’s bias score appears large simply because it performs well in many cases. This formula for measuring bias is too simplistic and does not appropriately account for the capabilities of large models, being influenced by their performance.
>
> - Thank you for your insightful question and for the opportunity to clarify our approach. The formula we proposed for quantifying self-preference bias is based on the difference in probabilities. Specifically, it calculates the difference between the recall of the LLM evaluator in high-quality cases (where ground truth  labels are available) and its recall in low-quality cases.
> - As you correctly pointed out, both the LLM evaluator and the LLM that generates responses are probabilistic models. To fully capture their stochastic nature, practical measures like conducting multiple sampling rounds may be meaningful. However, we believe that our proposed metric remains practical, as it measures the average bias tendency. The probabilistic variations in responses are effectively averaged out, preserving the metric's utility.
> - Regarding the potential impact of GPT-4’s considerably stronger performance on bias measurement, we do not expect it to be significant. Our formula considers probabilities within subsets evaluated as high-quality and low-quality, ensuring that the values are not influenced by the sample size of these subsets. Nevertheless, we acknowledge that if one subset has a smaller sample size, it could be more susceptible to noise from factors like annotator mislabeling or the stochastic behavior of the model.
>
>
> > Weakness 2: This method primarily focuses on specific conditions for bias, potentially overlooking other factors such as the content, context, and other semantic features of the input text. This narrow focus may lead to a partial understanding of the model's self-preference bias. Additionally, the scale of the analyzed data is insufficient and lacks comprehensiveness in broader contexts. The choice of specific input features in calculating the bias can influence the results.
>
> - We agree that the conditions of the input text are a crucial consideration. However, we believe that overlooking aspects of the content or context of the input text is more of a data-related issue rather than a methodological one. By incorporating input texts from diverse domains and tasks, our proposed method can effectively investigate self-preference bias in a more comprehensive manner. In cases where there is an imbalance in the number of samples for different conditions, if labels for each condition are available, we can analyze each condition separately by dividing them into subsets and calculating multiple scores.
> - For our experiments, we used the Chatbot Arena Dataset, which consists of 33,000 dialogue evaluations. Narrowing down to pairs that include a specific LLM evaluator reduced the sample size to approximately 500-4,000 instances, so achieving sufficient sample sizes for all LLMs remains a challenge. Nevertheless, we believe we utilized the largest publicly available dataset of dialogue evaluations.
> - Regarding domain and task diversity, the Chatbot Arena setup we used focuses on unconstrained conversational evaluations, without narrowing down to specific topics or tasks in our experiments. However, since Chatbot Arena is a platform accessible to anyone on the web for evaluation purposes, there is no guarantee of balanced category representation or comprehensive coverage, which is an acknowledged limitation.
>
>
> > Weakness 3:In the writing, Equation 3 lacks an explanation of ω.
>
> - We specify w under the summation symbol in Equation 3 to represent elements within the response set {A, B}. The term p(A|context) denotes the probability that the LLM evaluator outputs “A.” We used the summation notation to maintain consistency with the notation used by Schick et al. (2021), who employed same calculations.

---

> > ### Author Response · Authors · 2024-11-25
> >
> > > Question 1: Considering that LLMs may exhibit different biases depending on the context, how does your method account for contextual variations? Can the bias calculation be adapted to reflect dynamic changes in model behavior?
> >
> > - To measure the self-preference bias specific to a particular LLM across various contexts, we believe that using multiple contexts for a more detailed analysis is the most effective approach. This allows us to observe how bias varies with different contextual factors and better understand the dynamic changes in model behavior.
> > - By grouping contexts or using stratified sampling, we can refine our bias calculation to reflect these dynamic changes, offering a more adaptive understanding of bias in different situations.
> > - In practical terms, when evaluating an LLM as an evaluator within a fixed context, the entire setup, including the context, can be regarded as a single evaluation system. From this perspective, our method remains practical for measuring bias specific to that system.
> >
> >
> > > Question 2: How do you handle potential inconsistencies in human evaluations that serve as the benchmark for calculating bias? What processes are in place to mitigate subjectivity in these evaluations?
> >
> > - We believe the most effective approach to ensure the quality of human evaluators is to maintain diversity among them and implement rigorous quality control measures.
> > - Alternatively, another approach could be to utilize the Demographic Parity-based score we proposed in Section 6 (Discussion). This metric assesses how much higher an LLM evaluator rates its own outputs, independent of human labels. This allows for an analysis of scores in an extreme scenario where human subjectivity is completely removed.
> > - Furthermore, LLM-as-a-judge is a technique designed to serve as a substitute for human evaluators, and one of the uses of data created by LLM-as-a-Judge is to align LLMs with human values. Therefore, when measuring self-preference bias that considers discrepancies with human judgments, we recognize that incorporating human subjectivity is unavoidable.
> >
> >
> > > Question 3: Given your hypothesis that self-preference bias is related to text perplexity, how do you isolate the effect of perplexity from other factors when interpreting the results? What methodology do you use to analyze this relationship?
> >
> > - Thank you for your questions and comments. In our study, we first divided data into bins based on the level of perplexity and examined how the LLM’s evaluations varied within each bin. Our findings indicate that the LLM’s evaluations tend to fluctuate more significantly compared to human evaluations. This pattern was observed regardless of whether the response was self-generated or not. Additionally, we confirmed, as expected, that self-generated outputs had lower perplexity. These results suggest that self-preference bias may arise because self-generated outputs are associated with lower perplexity.
> > - We acknowledge that our current analysis primarily highlights the relationship between perplexity and self-preference bias, and we recognize the need to explore causal inference methods to rigorously establish causality.
> > - We also understand the reviewers’ concerns about other potential factors, such as response content, sentence length, and style, which may influence the evaluations. While decomposing these factors in detail would be highly beneficial, we believe that demonstrating the average impact of perplexity on evaluations is a crucial first step. Perplexity is an observable variable, making it relatively straightforward to control for in our analysis. Moving forward, we aim to explore a more comprehensive analysis of these factors to deepen our understanding of the bias.
> >
> >
> > > Question 4: Have you considered the impact of model capabilities and the data distribution of self-preference bias results obtained for each model on the bias?
> >
> > - First, we believe that the impact related to model capabilities is not heavily dependent on human evaluation recall. However, as mentioned earlier, if there is a significant imbalance between highly-rated and poorly-rated samples, it is true that the effect of noise can differ across these groups.
> > - We cannot claim that the influence of data bias is completely eliminated in our experimental setup. Since we are using the Chatbot Arena dataset, topics and tasks are not controlled. Nevertheless, it is possible to design experiments that minimize the influence of data distribution by isolating attributes and conducting analyses on specific subsets.

---

### Meta-Review · Area_Chair_51YN · 2024-12-17

**Metareview:**

The authors investigate the issue of self-preference bias when the same model is used to generate output and to judge it. They focus their analysis on perplexity, claiming that it is the fact that the lower perplexity of the model's own outputs is the cause of self-preference bias. The reviewers appreciate the authors' efforts in introducing a new metric for measuring self-preference bias in LLMs, their analysis across 8 LLMs, clarity of writing, and insights provided. However the reviewers also raise several concerns such as the simple nature of the formula to compute the metric that may fail to capture the dynamics of text generation or the fact that it doesn't properly take class imbalance and response length into account, and the overall novelty of the work. The authors attempt to address these concerns but overall their responses do not seem convincing; while promising, this work is not mature yet for publication.

**Additional Comments On Reviewer Discussion:**

The authors do a good job in addressing the reviewers' questions and mentioned weaknesses, particularly with respect to the formula of the proposed metric and details around evaluation setup and decisions. Only one reviewer responds (and is not convinced). I do appreciate the details provided in the author responses especially to reviewer rTG but overall the responses do not address the core concerns raised.

---

### Decision · Program_Chairs · 2025-01-22

Reject